# Stable Isotope Analyses Reveal Impact of Fe and Zn on Cd Uptake and Translocation by *Theobroma cacao*

**DOI:** 10.3390/plants13040551

**Published:** 2024-02-17

**Authors:** Rebekah E. T. Moore, Ihsan Ullah, Jim M. Dunwell, Mark Rehkämper

**Affiliations:** 1Department of Earth Science and Engineering, Imperial College, London SW7 2BP, UK; markrehk@imperial.ac.uk; 2School of Agriculture, Policy and Development, University of Reading, Reading RG6 6EU, UK; i.ullah@reading.ac.uk (I.U.); j.m.dunwell@reading.ac.uk (J.M.D.)

**Keywords:** cadmium, Cd, iron, Fe, zinc, Zn, *Theobroma cacao*, MC-ICP-MS, stable isotopes, hydroponics

## Abstract

High concentrations of toxic cadmium (Cd) in soils are problematic as the element accumulates in food crops such as rice and cacao. A mitigation strategy to minimise Cd accumulation is to enhance the competitive uptake of plant-essential metals. *Theobroma cacao* seedlings were grown hydroponically with added Cd. Eight different treatments were used, which included/excluded hydroponic or foliar zinc (Zn) and/or iron (Fe) for the final growth period. Analyses of Cd concentrations and natural stable isotope compositions by multiple collector ICP-MS were conducted. Cadmium uptake and translocation decreased when Fe was removed from the hydroponic solutions, while the application of foliar Zn-EDTA may enhance Cd translocation. No significant differences in isotope fractionation during uptake were found between treatments. Data from all treatments fit a single Cd isotope fractionation model associated with sequestration (seq) of isotopically light Cd in roots and unidirectional mobilisation (mob) of isotopically heavier Cd to the leaves (ε^114^Cd_seq-mob_ = −0.13‰). This result is in excellent agreement with data from an investigation of 19 genetically diverse cacao clones. The different Cd dynamics exhibited by the clones and seen in response to different Fe availability may be linked to similar physiological processes, such as the regulation of specific transporter proteins.

## 1. Introduction

Plants require a range of macronutrients (N, P, K, Mg, S, and Ca) and micronutrients (B, Cl, Mn, Fe, Ni, Cu, Zn, and Mo) to fulfil their physiological functions [1,2]. The homeostasis of these essential nutrients can be disrupted by low availability and non-specific uptake mechanisms, causing competition between metals for the same organic ligands [3,4]. For instance, transporters responsible for the uptake of Mn, Fe, and Zn [5,6,7], as well as Ca channels [8], can facilitate the uptake of toxic Cd. Such competition can provide opportunities to limit the uptake or translocation of toxic elements. The problems relating to the accumulation of Cd in crops, including *Theobroma cacao* (cacao), are well known [9,10,11,12,13,14], and a wide range of agronomic and genetic approaches [15,16,17,18] have been proposed to reduce the level of Cd to those considered acceptable by the various regulatory authorities [19].

One agronomic approach to interfere with the uptake and translocation of Cd in food crops is to supply Zn [20,21]. This is used because Zn and Cd are chemically similar, so they are likely to share uptake and translocation pathways. In detail, Zn and Cd may compete for binding sites (O, N, and S ligands) on biological molecules such as the ZIP (Zrt-, Irt-like protein) and ZnT (Zn transporter) family transporters responsible for root metal uptake and translocation [22,23,24]. Zinc fertiliser or foliar applications have been tested in attempts to reduce the overall accumulation of Cd in the harvested products in a range of crops. The majority of such studies have been conducted on *Oryza sativa* L. (rice) [25,26,27,28,29], specifically in response to the major problem of Cd-contaminated soils in China, which is largely due to rapid industrialisation [30]. Most studies found that the application of Zn to soils or directly to the leaves reduced Cd accumulation [25,26,27], while others found a dependency on soil pH [28] and elemental concentrations [29], where high Zn dosages may facilitate Cd accumulation. Increased soil Zn and well-timed foliar Zn treatments were found to reduce Cd uptake by *Triticum aestivum* L. (wheat), a monocotyledonous species like rice [28,31]. Similar trends were also found for dicotyledonous crops such as *Lactuca sativa* L. (lettuce) [32,33] and *Brassica chinensis* (pak choi) [34] in both pot and hydroponic experiments. Conversely, Cd and Zn uptake were synergistic in a hydroponic experiment with a different brassica species, *Brassica juncea* (Indian mustard) [35]. Similarly, Cd uptake increased with the addition of Zn to the nutrient solution in a pot experiment with (non-crop) woody dicotyledonous *Populus trichocarpa* (black cottonwood, or ‘western balsam-poplar’) [36]. In regard to cacao, a positive correlation has been found between Zn and Cd in leaves, as well as between the Zn in beans and Cd in leaves [37]. Soil and foliar Zn treatments have been studied both to overcome symptoms of Zn deficiency [38,39] and to mitigate against Cd toxicity [20]. It was found in a pot experiment that increasing soil Zn concentrations decreased both root uptake of Cd and translocation to the leaves [20].

Chemical relationships have also been observed between Cd and Fe in plants. Unlike Zn and Cd, Fe is often present in the non-bioavailable 3+ oxidation state. This is either reduced to Fe^2+^ before plant uptake or Fe^3+^ is bound by excreted phytosiderophores, which facilitate uptake [1]. It has been shown (e.g., for rice, wheat, and poplar) that low Fe availability enhances plant uptake of Cd [40,41], while high Fe levels in soils and in foliar treatments can reduce Cd uptake [42,43,44]. This is at least partly due to the regulation of Fe supply by the *IRT1* (Iron Regulation Transporter 1) gene, which has also been found to mediate the levels of Cd [1,45,46].

Natural stable isotope analyses provide more information on essential nutrient homeostasis in plants than is available from concentrations alone [47]. Differences in isotope compositions between soil or hydroponic solutions (source) and plants have provided new insights into uptake mechanisms [48,49,50,51,52]. Recent pot and hydroponic experiments investigated the stable isotope dynamics of Cd in the cereals rice [53,54,55,56], wheat, and barley (*Hordeum vulgare* L.) [52,57], in the Cd ‘accumulators’ cacao and castor bean (*Ricinus communis*) [51,58,59,60,61], and in the Cd hyper-accumulators black nightshade (*Solanum nigrum*) and *Sedum plumbizincicola* [60,61,62].

The Zn isotope dynamics in wheat, *S. plumbizincicola*, and cacao have also been investigated, providing further insights into the relationships between Zn and Cd [63,64,65]. For instance, for all three species, the plant roots preferentially take up isotopically light Zn *and* Cd, suggesting a shared uptake mechanism [63,65]. This is significant as it indicates that increased Zn supply could potentially be used to mitigate against high Cd uptake. For cacao, the magnitude of Zn isotope fractionation during uptake was slightly greater when there was more available Cd (20 μmol L^−1^ compared to 0 and 5 μmol L^−1^), but the overall Zn isotope dynamics were similar [63]. Analyses of roots versus above-ground parts for all three species revealed that the isotopes of the two elements fractionate in opposite directions once inside the plant, suggesting that the storage and translocation mechanisms are distinct for the two elements [63,64,65]. To a first order, the opposite isotope dynamics have been attributed to differences in the affinities of Zn and Cd isotopes to O, N, and S ligands in the roots and above-ground organs [66,67]. For rice, paired isotope and X-ray absorption spectroscopy (XAS) analyses found that the distinctively light Cd isotope compositions of the roots are indeed likely associated with S-ligand binding [53]. 

For the present study, six-week-old cacao seedlings, that had been grown under nutritionally sufficient hydroponic conditions, were transplanted to new solutions that featured high Cd concentrations. These solutions had the same concentrations of essential elements as before, but some did not contain any (excluded) Fe and/or Zn. Foliar solutions containing Fe or Zn were also applied to the leaves in some treatments. The controlled conditions enable the investigation of how phytoavailable Zn and Fe supply affects the concentration and stable isotope composition of Cd that is taken up, stored, and mobilised in the seedlings. An advantage of using hydroponics for such experiments, over field experiments with fruit bearing trees and ‘natural’ soils, is that whole-system mass inventories (solution, roots, and shoots) of (phytoavailable) metals can be readily and accurately obtained. The results enable an initial assessment of the importance of the two essential nutrients for the uptake and translocation of Cd. 

## 2. Materials and Methods

### 2.1. Plant Growth and Sample Preparation

This research complied with all relevant institutional, national, and international guidelines and legislation.

Seeds of cacao clone MO 20 were obtained from the International Cocoa Quarantine Centre (ICQC), Reading, UK. Seed coats were removed before planting in seed compost for germination. Two-week-old seedlings were transferred into nine 4 L plastic containers containing 3 L of half-strength Hoagland solution (Appendix A), adjusted to pH 5.2. Each container contained three seedlings. The nutrient solution was aerated for 15 min every two hours and renewed every week. 

Twenty-eight days after sowing, plants were moved to nine new freshly prepared half-strength Hoagland solutions. Of these nine ‘treatments’ (T), eight (T1-8) contained 20 μmol L^−1^ CdCl_2_ (equivalent to 2.25 mg L^−1^ Cd), and five treatments (T4-8) excluded either Fe (T4 and T5) or Zn (T6 and T7) or both (T8; Table 1 and Appendix A). The remaining container was a control (C), containing the half-strength Hoagland solution (including Fe and Zn) but no CdCl_2_. Where present, the concentrations of Fe and Zn in the hydroponic solutions were 2.38 mg L^−1^ (42.7 μmol L^−1^) and 25 μg L^−1^ (0.38 μmol L^−1^), respectively, and were administered as Fe-EDTA and ZnSO_4_. The Zn:Cd and Fe:Cd ratios are comparable with those in a previous hydroponic experiment [51,63]. Zinc concentration and isotope results showed that although the experiments were conducted at a much lower Zn:Cd than that found in bulk tropical soils, the Zn uptake and translocation dynamics were not substantially affected [63].

Four of the latter treatments also featured additional foliar applications of Fe (T2 and T5) or Zn (T3 and T7; Table 1 and Appendix A). For foliar applications, freshly prepared 0.1% solutions of Fe-EDTA (Fe 13 EDTA, Solufeed, Chichester, UK) and Zn-EDTA (Librel^®^ Zn, BASF, Ludwigshafen, Germany) were sprayed twice at one-week intervals on leaves, covering all surfaces on both sides of the leaves. Plants were cultured under controlled environment conditions (28/20 °C day/night temperature, 16 h photoperiod with 60% relative humidity). After 14 days of treatment, the plants were individually harvested and divided into leaves, stems, and roots. No seedlings showed any visible sign of Cd toxicity or mineral deficiency (necrosis or chlorosis; Appendix A).

The leaves and stems were washed with deionised water whereas the roots were submerged in 20 mmol L^−1^ Na_2_EDTA solution for 15 min to remove apoplastically bound Cd, followed by washing with deionised water. Plant material was oven-dried at 70 °C and ground using a rotor mill (Pulverisette 14, Fritsch). Initial Cd concentrations were obtained using inductively coupled plasma mass spectrometry (ICP-MS). For this, two 40 mg aliquots were taken from each leaf, stem, and root sample of every replicate plant (*n* = 3 per treatment). These were digested using 5 mL ultra-pure 70% HNO_3_ on a hotplate at 110 °C for 8 h.

### 2.2. Cadmium Concentration and Stable Isotope Analysis

Cadmium isotope composition measurements were undertaken in the Imperial College London MAGIC clean room and mass spectrometry laboratories. Distilled mineral acids, 30–32% Romil UpA^TM^ grade H_2_O_2_ and 18.2 MΩ cm H_2_O from a Millipore system, were used throughout sample preparation, which is described in detail elsewhere [51,68].

Two plants (or ‘biological replicates’) were randomly selected from each treatment for Cd stable isotope analysis. Hydroponically grown plants are less physiologically distinct from one another than plants grown in soils under controlled or field conditions. It has been demonstrated that for such systems, the stable isotope compositions and distributions of metals in the plants grown under the same experimental conditions are even more similar than their biomass and elemental concentrations [61,65,69]. The plants used for the current experiments were also clones, which further reduces the likelihood of physiological differences. Hence, two replicates were considered sufficient for effective statistically relevant data analysis. Furthermore, natural stable isotope data collected via multiple collector ICP-MS (MC-ICP-MS) are very precise, but they are much more time-consuming and expensive to obtain than concentration results via ICP-MS.

In brief, a further 10–80 mg aliquot (depending on the remaining sample mass) was taken from the leaf, stem, and root samples of each plant and digested using concentrated HNO_3_ and H_2_O_2_ in a Milestone Ethos EZ microwave system [51]. Aliquots of these digest solutions were then equilibrated with a suitable amount of ^111^Cd–^113^Cd double spike (DS) solution, based on the initial Cd concentrations obtained via ICP-MS, to create solutions with 200–300 ng Cd and ratios of spike-derived to natural Cd of between 1 and 2. Purified Cd fractions were then obtained using a three-step anion exchange and extraction chromatography procedure followed by liquid–liquid extraction [68]. For quality control, procedural blanks and concentration-certified reference materials (spinach leaf NIST SRM-1570a and white cabbage BCR-679) were processed alongside each sample batch. Even after the leaves, stems, and roots of the three control plant digests were pooled, there was insufficient Cd for reliable isotope measurements. Similarly, there was insufficient remaining sample for a precise isotope composition analysis for one of the T6 roots, so the third available plant from that treatment was used to ensure two full plants were robustly characterised isotopically for each treatment.

The purified Cd fractions were dissolved in 0.1 M HNO_3_ and run at Cd concentrations between 20 and 60 ng mL^−1^ using a Nu Instruments (Wrexham, UK) Nu Plasma HR or II MC-ICP-MS equipped with Faraday cups with 10^11^ Ω resistors and a Cetac Aridus II sample introduction system. Nebuliser uptake rate was ~0.11 mL min^−1^ and sensitivity for Cd was ~300 or ~650 V/(μg mL^−1^) on the Nu Plasma HR and II, respectively. To monitor instrumental drift, sample runs were bracketed by analyses of the NIST SRM-3108 Cd isotope reference material. A secondary pure Cd standard, BAM-I012, was also analysed for data quality control. Isotope compositions were calculated offline using double spike equations and an iterative procedure, while the final Cd concentrations were obtained with the isotope dilution technique using results from the double spike data reduction [70]. The latter were in good agreement with the preliminary concentrations measured via ICP-MS. Final cadmium isotope compositions are reported relative to the bracketing standard measurements of NIST SRM-3108 Cd [71]:(1)δC114d=C114d/C110dSampleC114d/C110dStandard−11000

When the isotope compositions of different Cd pools (i.e., the cacao roots, stems, and leaves, and CdCl_2_ in the hydroponic solution) are compared, the term is used to denote isotope fractionation:(2)$${\Delta^{114}Cd}_{A-B}=\delta^{114}{Cd}_{A}-\delta^{114}{Cd}_{B}$$

The total Cd masses in leaves, stems, and roots, calculated using the isotope dilution-derived Cd concentrations and dry biomass, were used to quantify root to shoot translocation (translocation factors, TFs):(3)$$TF(\%)=\frac{Cd_{leaf}+Cd_{stem}}{{Cd}_{root}}100$$

### 2.3. Statistical Analysis

A Kruskal–Wallis test was used to compare groups of data. Differences between datasets were considered as statistically significant if *p* < 0.05. This non-parametric test was used due to the small sample numbers and because most data did not fit a normal distribution, according to the Shapiro–Wilk test.

## 3. Results

### 3.1. Quality Control

Total plant dry biomass from treatments (T) 1–8 ranged from 0.57 g to 0.99 g, compared to 0.45 g to 0.99 g for the control plants. Leaves provided between 63% and 78% of the biomass (Appendix A). The ranges and standard deviations reveal that the variations seen for replicates of a single treatment (1sd = 0.01 to 0.21) are generally larger than the variance observed for the main biomass of the nine different treatments (1sd = 0.06). When the biomass of plants grown in treatments that contained Fe (*n* = 11) or Zn (*n* = 10) in the hydroponic solution are compared to those that did not (*n* = 6 and 7), there is no statistically significant difference (*p* = 0.06 and 0.49, respectively). There are insufficient data to apply such tests to plants with foliar treatments. In summary, this suggests that (i) the 20 μmol L^−1^ Cd treatment, and (ii) the different Fe and Zn treatments did not have a substantial impact on plant growth and development.

The Cd isotope compositions were not blank-corrected as all procedural blanks contained <100 pg Cd, which is equivalent to less than 0.1% of the Cd in even the least concentrated sample, and therefore negligible. The reported isotope data have excellent analytical precision (mean = 0.05 ± 0.03‰; 1sd), as determined from the 2sd of repeated runs of the bracketing primary standard during the measurement sessions. The Cd isotope compositions of the reference materials were identical, within analytical precision, to those reported in the literature [51,53,71,72,73]. In addition, there was good agreement between the measured Cd concentrations and the reference data for the two concentration-certified materials (Appendix A). 

### 3.2. Cd Speciation in the Hydroponic Solutions

Possible changes in Cd speciation due to the inclusion or exclusion of Fe and Zn in the hydroponic solutions were calculated using Visual MINTEQ ver. 3.1. software. These calculations show that for all eight treatments, Cd was mostly present in the solutions as Cd^2+^ (~88%) and CdSO_4_ (~6.2%), with only minor amounts present as CdHPO_4_, CdNO_3_^+^, Cd(SO_4_)_2_^2-^, and CdCl^+^ (0.07% to 3.6%) (Appendix A). When Fe-EDTA was present, only 0.7 to 1.4% of the Cd and 1.1% of the Zn were bound by EDTA. This compares to 99% of the Fe and about 60 to 76% of the Cu. These calculations show that the inclusion or exclusion of ZnSO_4_ and Fe-EDTA have a minimal effect on Cd speciation in the hydroponic solutions and are therefore unlikely to impact the uptake of Cd into the cacao seedlings. As such, any variation in Cd isotope compositions within the plants and between treatments can be ascribed to physiological processes that affected the Fe and Zn supply, rather than speciation effects and subsequent changes in Cd phytoavailability. 

### 3.3. Cadmium Concentrations and Isotope Compositions

The mean fraction of Cd that was taken up from the hydroponic solutions by the three plants of a treatment was 5% (ranging from 3 to 8%). The isotope compositions of the final solutions were calculated taking into account the amount of Cd taken up by the three plants (Table 1). 

The difference between the Cd isotope composition of the initial solution (δ^114^Cd_initial solution_ = 0.36 ± 0.04‰ 2sd, *n* = 6) and the calculated final solutions (δ^114^Cd_final solution_ = −0.33 to −0.35‰) was then used with the following Rayleigh equation to determine the isotope fractionation factor ε^114^Cd_uptake_ associated with plant uptake:(4)$$\delta^{114}{Cd}_{final\;solution}-\delta^{114}{Cd}_{initial\;solution}=\varepsilon^{114}{Cd}_{uptake}\ln\left(f\right),$$
where *f* is the fraction of Cd remaining in the final solution. Given that an average of only 5% was absorbed by the plants, adsorption of Cd to root surfaces was assumed to be considerably less significant, and similar across all treatments, such that it had negligible influence on δ^114^Cd_final solution_. The data and calculations yield a mean ε^114^Cd_uptake_ value of 0.29 ± 0.02‰ for the treatments, which is identical to the mean Δ^114^Cd_tot-sol_ value of 0.27 ± 0.03‰ (Table 2). The difference between the two results for each treatment (between −0.04‰ and 0.00‰) is smaller than the average analytical precision, indicating that reservoir effects due to the uptake of Cd from the solutions are negligible. Given this, and to be consistent with the nomenclature used in a previous investigation that grew cacao under similar conditions [51], the Δ^114^Cd_tot-sol_ values are used here in the following sections.

The treatments with hydroponically administered Zn + Fe have the highest total plant Cd (T1–T3), and the treatment with no added Zn or Fe (T8) has the lowest (Table 1). The plants given hydroponic Zn, but not Fe (T4 and T5), contain less Cd than those with Fe but no Zn (T6 and T7; Figure 1a). A similar pattern emerges when considering fractions of total Cd (*f*) in the three plant tissues (Figure 1b). In detail, treatments that administered Fe + Zn via hydroponics (T1–T3) yielded the largest Cd fractions in the leaves (*f*_leaf_) and those that excluded Fe and Zn (T8), _or only Fe (T4 and T5) have the lowest *f*_leaf_ (Figure 1 and Appendix A).

When Cd concentrations and isotope compositions are considered together, there is an overall trend of increasing δ^114^Cd with decreasing Cd concentrations [Cd]. This is primarily defined by differences between the three organs (Figure 2). The difference between root (R) and stem (S) isotope compositions tends to be small (mean of all treatments Δ^114^Cd_S-R_ = −0.03 ± 0.11‰, 1sd) but slightly larger differences are apparent for T3 and T7, both of which contain foliar Zn (mean Δ^114^Cd_S-R_ = −0.16‰ and −0.15‰, respectively; Table 2). The largest apparent within-plant isotope fractionations are usually recorded between leaves (L) and stems (mean Δ^114^Cd_L-S_ = 0.23 ± 0.05‰, 1sd; Table 2). The total plant Cd (tot) for all experiments was isotopically lighter than the CdCl_2_ (sol) added to all hydroponic solutions (Δ^114^Cd_tot-sol_ = −0.27 ± 0.03‰, 1sd; Figure 2 and Table 2).

There is a positive correlation between total plant Cd mass and the translocation factor (R^2^ = 0.77; Figure 3a). The data for plants that were administered the hydroponic treatments containing Fe + Zn or solely Fe almost ubiquitously populate the upper part of this trend (Cd > 100 ug, TF > 100%), while plants that were not given hydroponic Fe or Fe + Zn all have Cd < 100 µg and TF < 100%. This grouping is also observed when the apparent Cd isotope fractionation between shoots and total plants (Δ^114^Cd_shoot-tot_) is plotted against the fraction of total Cd in the shoots (*f*_shoot_; Figure 3b). In detail, the difference in Δ^114^Cd_shoot-tot_ between plants grown with (*n* = 10) or without (*n* = 6) hydroponically administered Fe is significant (*p* = 0.001), while no difference (*p* = 0.63) is found when considering treatments with (*n* = 10) or without Zn (*n* = 6). When the same variables are plotted but considering foliar treatments, the plants administered foliar Zn (*n* = 4) group together at high *f*_shoot_ and low Δ^114^Cd_shoot-tot_ (Appendix A).

## 4. Discussion

### 4.1. Effect of Fe and Zn on Cd Uptake

Due to multiple co-variables, such as the supply or limitation of Zn and/or Fe to the roots and/or leaves, quantitative comparisons between the eight individual treatments are of limited utility. More statistically robust and meaningful comparisons can be made when treatments with a shared variable are pooled. Using this approach, the Cd content data reveal the potential regulation of Cd uptake into cacao by Fe availability. In detail, a significant increase (with *p* = 0.006) in total Cd (or Cd uptake) is observed when hydroponic treatments that contained Fe (*n* = 10) are compared to those that did not (*n* = 6), while no such difference is apparent if the same comparison is conducted for Zn (*p* = 0.23) (Table 1 and Figure 1). Considering this, the similar Cd results obtained for the treatments that administered only Zn via the hydroponic solution or which excluded both Fe and Zn (Table 1 and Figure 1) indicate that the lower Cd uptake of these treatments is more likely associated with Fe limitation, rather than Zn supply. 

An important consideration is that when plants take up Fe in the form of EDTA complexes, the EDTA can then be released to the nutrient solution and potentially chelate other metals. Chelation of Cd would induce isotope fractionation between different Cd species in the solution, with heavy isotopes enriched in Cd-EDTA complexes [67]. This would decrease the amount of phytoavailable free Cd^2+^ ions in the solution, and these would have a lighter isotope composition. The absence of any significant difference (*p* = 0.59) in isotope fractionation on uptake between treatments with Fe-EDTA (*n* = 10) or without Fe-EDTA (*n* = 6), and indeed between all treatments (Δ^114^Cd_tot-sol_ = −0.27 ± 0.03‰, 1sd; Table 2), indicates that any difference in Cd speciation between the treatments did not substantially affect the mechanism by which the Cd was taken up. This implies that the chelation of Cd by EDTA was of such limited extent so as not to induce any measurable change to Cd uptake processes. Likewise, the consistent Cd isotope fractionation on uptake suggests that the uptake mechanism(s) remains the same. Furthermore, the Cd concentration and isotope data do not reveal any changes in Cd uptake resulting from the application of foliar Fe-EDTA or Zn-EDTA. This suggests that foliar Fe and Zn treatments do not impact Cd uptake, but this should be corroborated with more analyses.

There is not one widely accepted systematic relationship between the uptake of Zn and Cd by various plants. This is shown by Zn fertilisation experiments, which revealed increases [29,36], decreases [25,28,29,33], and no change [28,74,75] in plant Cd uptake due to Zn addition. For instance, Zn fertilisation on paddy fields is not always an effective Cd remediation strategy for rice, whereas the opposite has been presented for wheat [28]. Notably, the new hydroponic data for cacao presented here appear to differ from the results of a cacao pot study, which showed that additional Zn fertilisation decreased Cd uptake in seedlings [20]. The two studies differ in that they monitor Cd when Zn is either removed or supplemented in the root nutrient source. The apparent discrepancy, therefore, may reflect differences in the dosages and chemical speciation of Cd and Zn in the respective nutrient sources (hydroponics versus soil). Differences in gene expression between the two distinct cacao varieties that were employed in the studies may also impact the Cd-Zn uptake dynamics [18,51]. The MO 20 cultivar of this investigation is originally a Peruvian clone of the Forastero group, while the CCN 51 plants of the pot study represent a well-characterised variety from Ecuador, which has Iquitos, Criollo, and Amelonado group ancestry, and is commonly used in breeding programmes [20,76].

An alternative explanation for the lack of significant change in Cd uptake between the treatments of this study with and without Zn (*p* = 0.33) is that the change in the Cd to Zn ratio was insufficient to alter the mechanisms handling both elements. To illustrate this hypothesis, the example of high- or low-affinity transport proteins with low element specificity can be used. When certain nutrients (e.g., Zn or Fe) are in low/high supply, the expression of genes that encode non-specific proteins may change to regulate the uptake of the nutrients. In summary, the competition between Cd and Zn may have been too high for all treatments to induce any observable physiological effects from the exclusion of Zn.

The synergistic relationship between Fe and Cd uptake by cacao in Fe-deficient and -sufficient conditions is not seen for soil–plant studies of other species with different Fe availabilities. For wheat and rice, the uptake of Cd was enhanced in Fe-deficient conditions [41], and unchanged or reduced by fertilisation with dissolved and nanoparticulate Fe, under iron-sufficient conditions [43,44,74]. Foliar Fe treatments were, furthermore, reported to reduce Cd uptake by rice [42]. This antagonistic relationship has also been observed for phytoplankton, where Fe limitation in surface waters was shown to correlate with an enhanced uptake of Cd [77], possibly due to the regulation of high-affinity transport proteins. However, the environmental conditions and biological processing in seawater plankton and the hydroponically grown plants of this study are distinct, so the same uptake dynamics are not necessarily expected. The distinct Fe-Cd synergy observed in the current study for cacao may also be difficult to replicate in soils, which can exhibit a low phytoavailability of Fe, but the complete exclusion of Fe is not possible [48,78]. Alternatively, it may be a consequence of physiological differences in the way different plant species and plant types (e.g mono- vs. dicotyledonous species) take up and store both elements. The synergy with Fe observed here for cacao may hence be related to the distinct ability of this species to take up and tolerate high concentrations of Cd, and for which it may arguably be called an ‘accumulator’ plant. 

The consistent negative isotopic fractionation between total plant Cd and the solutions (Δ^114^Cd_tot-sol_ = −0.27 ± 0.03‰, 1sd, *n* = 16)_,_ implies that cacao roots take up light Cd isotopes preferentially, in excellent agreement with the observations of a previous hydroponic cacao study, which found Δ^114^Cd_tot-sol_ = −0.22 ± 0.08‰ (1sd, *n* = 20) [51]. This similar magnitude of fractionation in all treatments suggests that while total Cd uptake can differ between cacao plants grown under different conditions, they appear to use the same dominant uptake mechanism(s). This may be associated with Fe and Cd sharing common membrane transporter proteins [41,45,79,80]. The expression of a cacao gene (*TcNRAMP5*) that encodes one such protein, NRAMP5, has been found to be reduced under Fe-sufficient conditions and Cd supplementation, compared to nutrient-deficient and low-Cd conditions [80]. Furthermore, transgenic yeast expressing *TcNRAMP5*, which is mostly expressed in cacao roots, takes up more, and preferentially isotopically lighter, Cd compared to yeast modified with the empty vector [51,80]. Conversely, no change in the expression of *TcNRAMP5* was reported in cacao seedlings that were grown in Zn-deficient conditions [80], which supports the hypothesis of a possible decoupling between the processing of Cd and Fe, and Zn in cacao.

However, in apparent contrast to the findings summarised above, Cd taken up by modified rice with a knock-out of the *OsNRAMP5* gene was isotopically lighter in comparison to the Cd taken up by the wild type [54]. This suggests that the preferential uptake of light Cd isotopes, as observed for rice at least, may be dominated by mechanisms other than membrane transport via NRAMP5. However, understanding the importance of individual plant genes is complex, and the reduction of *OsNRAMP5* expression may have promoted the upregulation of such alternative mechanisms, and this may not be representative for non-mutant conditions. A particular challenge is relating the expression of a gene to the level of protein encoded by that gene [81]. This follows from several considerations. First, it is known that many external stresses, including mineral nutrient exposure [82,83], increase the frequency of alternative transcripts produced from any gene, and this may lead to truncated transcripts and equivalent changes in the translated protein. Second, the protein levels depend on the coordinated regulation of the turnover of any specific transcript and on the encoded protein. In addition, environmental stress may affect the post-translational characteristics of the protein, thus possibly altering its interaction with other proteins, and its localisation within the cell. Although various technologies have been applied in this field [84], there is still a lack of comprehensive studies that relate the transcriptome and proteome during the response of cells to Cd exposure. Notably, such analyses are particularly difficult for membrane transporters such as NRAMP5 [85]. A combination of transgenic and gene-editing experiments would be useful to unravel the roles of individual membrane transport proteins in Cd uptake in cacao, and to determine to what extent they are impacted by different Zn and Fe availability.

### 4.2. Effect of Fe and Zn Treatments on Cd Translocation

Comparison of the leaf and root Cd isotope data shows that lighter isotopes are preferentially stored in the roots, while heavier isotopes are translocated (Figure 2). This is in agreement with the results of a previous hydroponic investigation of cacao that measured the Cd isotope compositions of leaves and roots from 19 genetically diverse clones [51]. For Zn, in contrast, cacao roots were found to be isotopically heavier than the shoots [63].

While the lowest total plant Cd contents, and therefore the lowest Cd uptake, was found for treatments that did not contain Fe in the hydroponic solutions (T4, T5, and T8), the low *f*_leaf_ values (of ~0.2) and Cd TFs (<100%) associated with these treatments suggest that Fe limitation also increases the root sequestration of Cd (Figure 1a and Figure 3a, and Appendix A), and vice versa. The apparent impact of Fe supply on the uptake and translocation of Cd (Figure 3a) is further corroborated by the Cd isotope data, as plants from hydroponic treatments that contain Fe consistently had lower Δ^114^Cd_shoot-tot_ values than those without (Figure 3b). This indicates that the addition of Fe to the nutrient solutions was associated with less Cd sequestration in roots, and thus more mobilisation of isotopically light Cd to the shoots.

The exponential relationship of Figure 4 is indicative of Rayleigh fractionation in a closed system, with unidirectional transport of isotopically heavy Cd to the leaves. Equation (5) describes the relationship between the sequestration and mobilisation of Cd in such a system:(5)$$\delta^{114}{Cd}_{mob}=\delta^{114}{Cd}_{tot}+\varepsilon^{114}Cd_{seq-mob}\ln\left(f_{mob}\right),$$
where δ^114^Cd_mob_ and δ^114^Cd_tot_ denote the Cd isotope compositions of the mobilised Cd and the total plant, respectively; *f*_mob_ represents the mass fraction of mobilised Cd relative to Cd_tot_; and ε^114^Cd_seq-mob_ is the isotope fractionation factor between sequestered (seq) and mobilised (mob) Cd. The δ^114^Cd_shoot_ or δ^114^Cd_leaf_ values are used for the δ^114^Cd_mob_ portion, as the shoots and leaves are the Cd ‘sinks’ and the total plant is the ‘source’. The model was forced through y = 0 at f_sink_ = 1 since such compositions must, by definition, have Δ^114^Cd_sink-source_ = 0‰. 

Single Rayleigh fractionation models were previously fitted to Cd isotope datasets from wheat [52], rice [53], and Cd accumulator species [61], and these yielded different extents of isotope fractionation with distinct ε^114^Cd_seq-mob_ values. These studies applied two or more distinct culturing conditions with different amounts of Cd and/or Cd in media with different chemical speciation. Together, these observations have two main implications. First, all studied plant species preferentially sequester isotopically light Cd in roots, regardless of growth media Cd concentration or speciation, leading to translocation of isotopically heavy Cd to the shoots. Second, the dominant mechanism(s) responsible for the sequestration and mobilisation of Cd appear to be distinct between species.

A single Rayleigh fractionation curve was shown to account for the data previously acquired for 19 genetically diverse cacao clones, with a ε^114^Cd_seq-mob_ fractionation factor, calculated using the Δ^114^Cd_leaf-tot_ values of each clone, of −0.13‰ [51]. The current dataset, which assesses Cd translocation in only a single clone (MO 20) but for different treatments, yields an identical overall ε^114^Cd_seq-mob_ value of −0.13‰ for Δ^114^Cd_leaf-tot_, and of −0.09‰ for Δ^114^Cd_shoot-tot_ (Figure 4). When calculated individually for each plant, the results yield mean ε^114^Cd_seq-mob_ values of −0.12 ± 0.02‰ (Table 2) and −0.07 ± 0.07‰ (*n* = 16), when Δ^114^Cd_leaf-tot_ and Δ^114^Cd_shoot-tot_ are used, respectively. It was suggested that the results for the 19 cacao clones define a single fractionation curve because they all utilise the same sequestration and translocation mechanisms, but with varying efficiency [51]. In the context of the current study, this may imply that the availability of Fe impacts the efficiency of the molecular mechanisms of Cd sequestration and translocation in a manner similar to that observed for different cacao clones reared in identical conditions. 

In detail, it is likely that a set of active mechanisms (e.g., involving transporters), rather than passive diffusion and transpiration pull, control the translocation of Cd in cacao and that Fe availability also regulates these mechanisms. The data suggest that Cd can trigger these transport mechanisms alone, and that they are upregulated when Fe is also readily available for uptake. Given this, Cd can be considered a successful ‘hitchhiker’ element, with translocation pathways that are not impeded by increased concentrations of the elements it ‘competes’ with for transporters. This synergism between the translocation of two metals was previously observed for Zn and Cd in *P. trichocarpa,* a species that has been compared to hyperaccumulators, as it can withstand high Zn and Cd concentrations in its shoots and leaves before toxic effects are noticeable [36]. Since there is no precedent for such dynamics with Fe and Cd, a specific mechanism cannot be hypothesised at this stage, and further experiments with a full-factorial approach are needed to better characterise the synergy, before specific processes are investigated.

A recent study found that the 19 cacao clones that were grown under the same condition and previously investigated for Cd isotopes also define similar ε^66^Zn_seq-mob_ values. In accord with the results for Cd, this suggests that they also utilise similar Zn sequestration and mobilisation mechanisms [63]. However, the ε^66^Zn_seq-mob_ fractionation factor was found to be significantly lower at low-Cd conditions, when only 0 or 5 μmol L^−1^ of Cd were added to the hydroponic solutions, as opposed to 20 μmol L^−1^. This indicates that the plants may use distinct mechanisms for the storage and transport of Zn when Cd is in low and high supply. In the current investigation, there is no difference in the ε^114^Cd_seq-mob_ values of plants grown in treatments with and without Zn, as both yield ε^114^Cd_seq-mob_ = −0.13‰ (*n* = 10 and 6, *p* = 0.66 and 0.28 when leaf and shoot are considered the sink, respectively). The lack of such differences in the current study suggests that Cd differs from Zn in that the mechanisms which control Cd sequestration and mobilisation are not altered by changes in the Zn supply. However, experiments with different concentrations of both elements are required to verify this conclusion.

While Fe availability appears to dominate the changes in Cd translocation in the current experiments, it is of note that treatments which involved foliar Zn-EDTA featured amongst the highest *f*_shoot_ values coupled with the lowest Δ^114^Cd_shoot-tot_ and Δ^114^Cd_stem-root_ (Appendix A). This may be indicative of a mechanism that promotes Cd translocation, thereby reducing the Cd fraction sequestered in the roots, when foliar Zn is administered. This tentative interpretation differs from the results obtained for monocotyledonous species, such as rice [26,27] and wheat [31]. This warrants further investigation, especially on more mature cacao plants, where foliar fertilisation may have a more significant impact on (re-)translocation of metals [31]. 

The biological or physicochemical mechanisms that control the preferential sequestration of light Cd isotopes in the cacao roots could relate to processes that induce isotope fractionation during membrane transport and/or to bonding with ligand donors of organic complexes. A mutant wild-type experiment on rice found that overexpression of *OsHMA3* resulted in heavier Cd isotopes in the shoots, signifying an increased retention of light Cd isotopes in the root vacuoles [54]. An XAS study found that Cd was predominantly bound to thiol groups (S-ligands) in the roots of rice with a functioning *OsHMA3* allele, whereas there was a 10–15% increase in Cd bound to carboxylate groups (O-ligands) in rice without the allele [86]. These findings, however, do not necessarily imply that HMA3 proteins bind Cd with S-donor ligands. In addition, membrane transport itself might not be responsible for the distinct ε^114^Cd_seq-mob_ value that appears to be characteristic for cacao. An alternative hypothesis is that increased activity of HMA3 allows more Cd to be sequestered and this sequestered pool may be complexed by glutathione, metallothioneins, or phytochelatins, possibly as part of the plant’s Cd detoxification strategy [87]. As this strategy would involve the bonding of Cd with thiol groups (with Cd-S bonds), it would explain the XAS results whilst also inducing an enrichment of light isotopes in the roots [67]. Further focused experiments on membrane proteins and chelating molecules are required to disentangle the processes responsible for the distinct isotope fractionation.

The translocation of Cd from roots to shoots may be modified by the impact that Fe availability has on sequestration and mobilisation mechanisms. This could include modifying the expression of genes that encode tonoplast transport proteins, which are in turn likely regulated by diverse chemical signals within the plant. For instance, in rice, *OsHMA2*, which encodes a protein that mediates root to shoot transport of Zn and Cd [88], was shown to be upregulated under Zn-limiting conditions [89]. Further research is necessary to determine whether the synergistic relationship between Cd and Fe can be reproduced in experiments with different availabilities of both metals and in mature cacao trees, especially for Cd translocation to the beans.

## 5. Conclusions

Analyses of Cd concentrations and isotope compositions in cacao seedlings from hydroponic experiments show that uptake and translocation of Cd were lowest when the plants were in hydroponic solutions without Fe, whilst removal of hydroponic Zn appeared to have little to no impact. This assessment is supported by a Rayleigh fractionation model, which suggests that root sequestration of Cd was enhanced when Fe was not present in the solutions. The isotope fractionation factor ε^114^Cd_seq-mob_ that was observed for Cd translocation was identical to previous results obtained in a similar hydroponic study of different cacao clones. This implies that the root sequestration and/or translocation mechanisms of cacao, which are the same among different clones but active to different extents, are also affected by changing Fe dynamics. Whilst uptake and translocation of Cd did not appear to be impacted by application of foliar Fe-EDTA, the data tentatively indicate that foliar Zn-EDTA may increase Cd translocation, as shown by lower Δ^114^Cd_shoot-tot_ fractionations and high *f*_shoot_ values, compared to other treatments. 

To further constrain the effect of Zn and Fe fertilisation on Cd uptake by cacao, experiments that apply variable concentrations of hydroponically administered Zn, Fe, and Cd are desirable. Furthermore, the concentrations and isotope compositions of all three elements should be measured. Such experiments would help to constrain any mechanistic relationships and to assess whether the lack of significant changes in Cd uptake and translocation with Zn availability is due to concentration effects or other factors such as genetics or elemental speciation. Incremental sampling of hydroponic solutions would help to better constrain the effect of speciation and uptake rate on Cd isotope fractionation during uptake. Such experiments could also be conducted with transgenic yeasts or model plants encoded with cacao membrane transporter genes to understand how individual proteins are affected. Furthermore, isotopic studies of wild-type and mutant cacao, which target specific transporters such as NRAMP5 and IRT-1, would further our understanding of the physiological mechanisms responsible for Cd uptake. Likewise, other transporters, such as HMA3 and HMA2, could be investigated to enhance our understanding of the specific processes associated with the isotope fractionation of Cd during sequestration and translocation from roots to shoots. 

## Figures and Tables

**Figure 1 plants-13-00551-f001:**
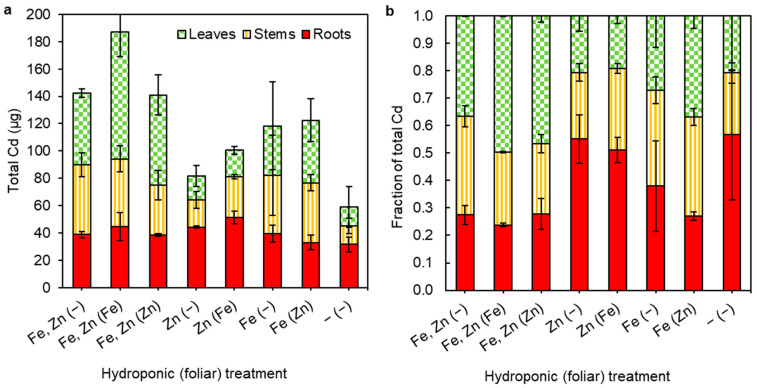
(**a**) Total Cd and (**b**) fraction of total plant Cd in roots, stems, and leaves for each treatment, which are described on the x-axis, where open and (bracketed) symbols represent the presence of Fe and/or Zn in the hydroponic solutions or foliar sprays, respectively. Values are averages from two biological replicates with 1sd error bars.

**Figure 2 plants-13-00551-f002:**
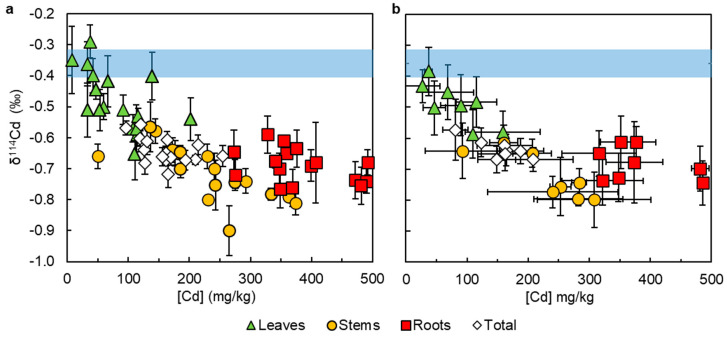
Cadmium isotope compositions (as δ^114^Cd) versus concentrations for cacao leaves, stems, roots, and total plants. Blue bar = isotope composition of the CdCl_2_ added to each hydroponic nutrient solution (δ^114^Cd = −0.36 ± 0.04‰). (**a**) All datapoints, where error bars represent the 2sd of repeat analyses of the primary Cd isotope standard that was run alongside the samples; (**b**) average values for each treatment, where the isotope compositions were calculated using inverse-variance weighting.

**Figure 3 plants-13-00551-f003:**
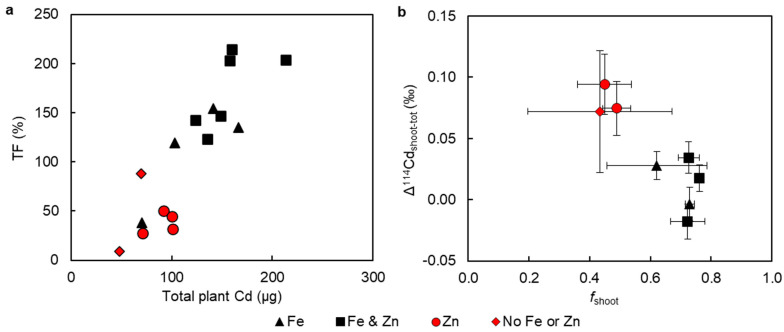
(**a**) Cd translocation factor (TF) versus total plant Cd, R^2^ = 0.77; (**b**) apparent Cd isotope fractionation between Cd in shoots and total plants versus fraction (*f*) of total plant Cd in shoots (values are averages of two replicates for each treatment, with average isotope compositions calculated using inverse-variance weighting). Both plots show which elements (Fe, Zn) were in the hydroponic nutrient solutions.

**Figure 4 plants-13-00551-f004:**
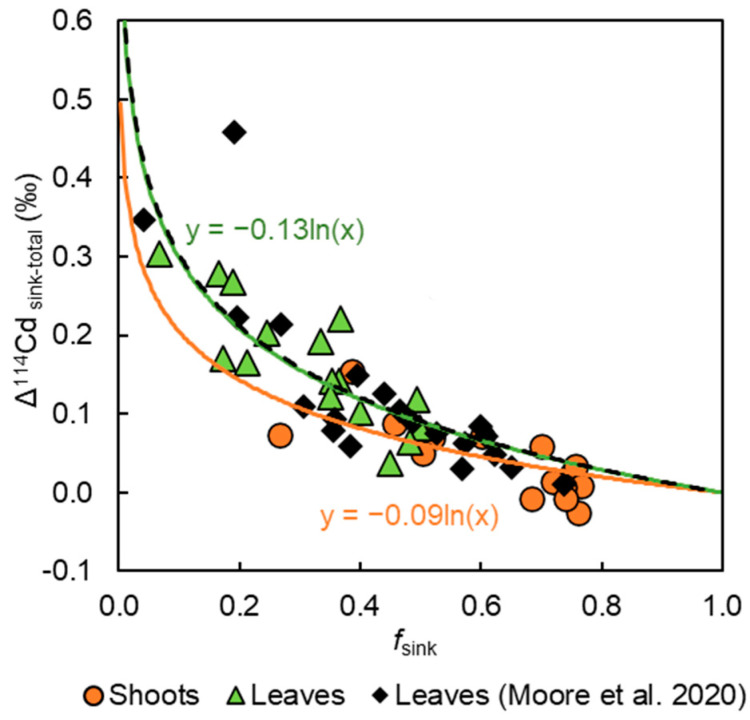
Apparent isotope fractionation between Cd sink (leaves or shoots) and total plant Cd plotted versus fraction of total plant Cd in sinks. The solid green, orange, and black dotted lines denote the fitted Rayleigh fractionation models for the leaves and shoots of this study and the leaves from Moore et al. 2020 [51], respectively. The ε^114^Cd_seq−mob_ values for both leaf datasets and the shoots are −0.13‰ and −0.09‰, respectively.

**Table 1 plants-13-00551-t001:** Cadmium concentrations and isotope compositions of plant tissues and total plant Cd for the eight treatments.

T	Nutrient Solutions		[Cd]			Cd	δ^114^Cd			
			(mg/kg, ± sd)			(μg, ± sd)	(‰, ± 2sd)			
	Hydroponic	Foliar	Roots	Stems	Leaves	Total	Roots	Stems	Leaves	Total
1	Fe, Zn	-	322 ± 67	284 ± 12	115 ± 33	142 ± 9	−0.74 ± 0.08	−0.74 ± 0.04	−0.48 ± 0.08	−0.64 ± 0.04
2	Fe, Zn	Fe	348 ± 1	282 ± 73	159 ± 62	187 ± 38	−0.73 ± 0.08	−0.80 ± 0.02	−0.58 ± 0.07	−0.67 ± 0.04
3	Fe, Zn	Zn	352 ± 34	253 ± 17	109 ± 2	141 ± 24	−0.61 ± 0.08	−0.76 ± 0.09	−0.59 ± 0.08	−0.66 ± 0.05
4	Zn	-	482 ± 14	160 ± 23	37 ± 6	82 ± 14	−0.70 ± 0.07	−0.61 ± 0.06	−0.38 ± 0.08	−0.62 ± 0.05
5	Zn	Fe	486 ± 6	207 ± 31	47 ± 19	101 ± 0	−0.74 ± 0.07	−0.65 ± 0.04	−0.50 ± 0.09	−0.67 ± 0.04
6 *	Fe	-	377 ± 32	240 ± 106	69 ± 42	118 ± 68	−0.61 ± 0.05	−0.77 ± 0.05	−0.45 ± 0.09	−0.63 ± 0.04
7	Fe	Zn	316 ± 61	308 ± 93	90 ± 33	122 ± 27	−0.65 ± 0.07	−0.80 ± 0.09	−0.49 ± 0.10	−0.64 ± 0.05
8	-	-	374 ± 47	93 ± 61	27 ± 28	59 ± 15	−0.68 ± 0.13	−0.64 ± 0.09	−0.43 ± 0.05	−0.61 ± 0.10
		**Mean**	**382**	**228**	**82**	**119**	**−0.68**	**−0.72**	**−0.49**	**−0.64**
		sd	66	72	45	40	0.05	0.07	0.07	0.02
		Min	316	93	27	59	−0.74	−0.80	−0.59	−0.67
		Max	486	308	159	187	−0.61	−0.61	−0.38	−0.61

T = Treatment. After an initial 28 days in the half-strength Hoagland solution (pH 5.2) without CdCl_2_, the cacao seedlings were grown for 14 days in eight different solutions, that all contained 2.25 mg L^−1^ Cd (20 μmol L^−1^ CdCl_2_). These solutions included or excluded Fe and Zn. Where present, the concentrations of Fe and Zn in the hydroponic solutions were 2.38 mg L^−1^ and 25 μg L^−1^, respectively, and were administered as Fe-EDTA and ZnSO_4_. The foliar treatments were sprays of 0.1% solutions of Fe- and Zn-EDTA. Values are the average of two (* or three) biological replicates, and isotope composition averages were calculated using inverse-variance weighting.

**Table 2 plants-13-00551-t002:** Cadmium isotope fractionation between different plant parts and total plants.

T	Nutrient Solutions		Δ^114^Cd (‰)						ε^114^Cd (‰)
	Hydroponic	Foliar	L-S	S-R	L-R	Tot-Sol	L-Tot	Sh-Tot	Seq-Mob
1	Fe, Zn	-	0.28	0.00	0.25	−0.29	0.17	0.03	−0.15
2	Fe, Zn	Fe	0.22	−0.07	0.14	−0.31	0.09	0.02	−0.12
3	Fe, Zn	Zn	0.16	−0.16	0.01	−0.28	0.04	−0.02	−0.10
4	Zn	-	0.23	0.07	0.31	−0.25	0.23	0.09	−0.15
5	Zn	Fe	0.14	0.09	0.24	−0.31	0.17	0.07	−0.10
6	Fe	-	0.27	−0.09	0.19	−0.26	0.16	0.03	−0.15
7	Fe	Zn	0.27	−0.15	0.14	−0.29	0.13	0.00	−0.11
8	-	-	0.27	0.09	0.23	−0.21	0.13	0.07	−0.11
		**Mean**	**0.23**	**−0.03**	**0.19**	**−0.27**	**0.14**	**0.04**	**−0.12**
		sd	0.05	0.11	0.09	0.03	0.05	0.04	0.02

Hydroponic solutions were half-strength Hoagland (pH 5.2) and all contained 20 μmol L^−1^ Cd. T = treatment; L = leaf; R = root; S = stem; Tot = total plant; Sol = Cd added to hydroponic solutions (δ^114^Cd = −0.36 ± 0.04‰); Sh = shoot (stem + leaf); Seq-Mob values calculated using leaf as the sink and total plant as the source. Values are averages of two replicate plants per treatment that were calculated using inverse-variance weighting. The mean and sd are calculated from the data in the table (*n* = 8), not individual replicate values.

## Data Availability

All data generated or analysed during this study are included in this published article (and its Appendix A).

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
