# Peer review of "Stable Isotope Analyses Reveal Impact of Fe and Zn on Cd Uptake and Translocation by Theobroma cacao"

_plants, 2024, doi:10.3390/plants13040551_

Round 1
Reviewer 1 Report
Comments and Suggestions for Authors
The manuscript "Stable Isotope Analyses Reveal Impact of Fe and Zn on Cd Up take and Translocation by Theobroma cacao" identifies the relation between Cd uptake and Fe and Zn availability. This is an extensive study and and very well presented data by authors.
Reviewer 2 Report
Comments and Suggestions for Authors
The manuscript reflects actual issues. While several studies have been conducted in field conditions, presented experiments are conducted in hydroponic conditions. The aim was to determine the influence of Fe and Zn intake and Cd translocation in different cocoa tissues. The theoretical introduction is relatively superficial and does not present a focused and clear basis for given experimental setup and interpretations. This section would benefit from focusing on literature data in clear directions (effect of Fe on Cd intake, Zn on Cd intake, effect of foliar application vs soil application, etc.). It is necessary to include information on ion competitive relationships and known transport mechanisms for Cd, for which the transport of Fe and Zn is relevant.
Methodologies do not clearly state how many biological replicates were evaluated in individual analyses (only mentioned in methods: "the plants were individually harvested" and "Two 40 mg aliquots were taken from each sample" – line 130). From the tables, it is evident that authors sometimes worked with 2, other times 3 samples of a variant; this information is crucial for statistical and reliable interpretation.
Authors worked with variants where the growth medium contained Cd. They related its accumulation (total or isotopic fractions) in different types of tissues to the addition of Fe and/or Zn in the growth medium (except according to table S4 variant 8, without additional Fe or Zn); the number of variants in this regard is unbalanced. Fe or Zn was applied foliarly, but not both metals together. In my opinion, this setup represents an imbalance limiting statistical assessments of individual factor impacts. Authors used non-parametric tests due to small sample numbers and data not fitting a normal distribution; these analyses, however, do not provide clear statistical significance of differences between treatments. In my opinion, 2 biological replicates are insufficient for reliable analyses of this experimental setup. I do not understand the rationale for averaging Cd content measured in different plant tissues, nor the averaging of data in individual tissues with the application of Fe and/or Zn foliarly or in the medium (data in tables). Values in figure 1 visualize differences in Cd accumulation in various variants or tissue types clearly but only from 2 biological replicates.
The second important and not entirely clear aspect of the manuscript is the description of the experiment/data regarding applied metals. Hoagland's solution contains micronutrients, including Fe and Zn. Therefore, I suggest consistently emphasizing/editing the expression "excluding" for relevant variants, for example, to "addition/no addition of Fe or Zn." Similarly, "either included sufficient amounts of Fe and/or Zn or omitted the two elements" – what exactly do "sufficient" and "omitted" represent? Expression such as „presence or absence of Fe and Zn in hydroponic solutions“ (line 201) – does it refer to the metal content of the medium itself or the treatment? Does the content of Fe and Zn in the half-strength medium represent insufficient concentration for plants? Authors should clearly define terms in relation to available/delivered metal concentrations in the introduction and use them consistently in the manuscript.
From the current form of the manuscript, it is not possible to attribute observed changes clearly to a specific treatment (I suggest conducting post hoc tests), reflecting unclear interpretations like "the lower Cd uptake of these treatments is associated with Fe limitation, rather than Zn supply."; "there is an overall trend of increasing δ114Cd with decreasing Cd concentrations [Cd]", the sentence in lines 266-269, and others.
Authors claim: "the variable availability of Fe impacts the efficiency of the molecular mechanisms of Cd sequestration and translocation in a manner similar to that observed for different cacao clones " (lines 449-451) – what does "variable" mean: range of deficiency to excess? Does Fe have any effect on Cd uptake and translocation? Similarly unclear is the statement: "Cd differs from Zn in that the mechanisms which control Cd sequestration and mobilisation are not altered by changes in the Zn supply." (line 462).
In evaluating Cd isotope translocation within the plant, transpiration plays a role. How does the application of Zn or Fe interfere with this factor when applied in hydroponics, foliarly and in combination of these treatments?
The above unclarities distort the value of obtained data, so the manuscript needs significant revision to eliminate unnecessary ambiguities and reflect more clearly on new findings. In figure 1, I suggest modifying the names of variants on the x-axis (it is not clear what the labels in parentheses or dashes mean. If it is a foliar treatment, the label should be part of the figure legend, not the text in the figure.
Reviewer 3 Report
Comments and Suggestions for Authors
The manuscript analyses the Cd content and isotopic fractionation in different tissues of T. cacao plants, grown for 2 weeks in hydroponic Cd solutions containing or not Fe and Zn, and/or with foliar application of those essential nutrients. The aim of the work was to assess the importance of the two essential nutrients in the uptake and translocation of Cd, as its level in the final product –chocolate- is regulated in a number of markets.
The experimental design is correct and the determinations have been very clearly described. Results and calculus are informative and interesting for the readers. I only have a few questions and suggestions, being the most important one to provide a possible explanation for the increase in Cd uptake when Fe is present in the hydroponic solution, as most of the evidence in the literature -in other species-, found the opposite trend:
Line 48
which are the main tendencies or conclusion in the cited studies? It is particularly important to mention the behaviour of dicotyledoneous species like T. cacao.
Line 319
As pointed out in the introduction, the most relevant study to compare with is the one performed in lettuce [33]. Uptake mechanisms usually differ between monocots and dicots
Line 364
Although explanation provided in lines 364-367 is consistent with the isotopic fractionation results, it seems to contradict what was found in the different nutrient treatments. If Cd and Fe share the same uptake transporter/s and under Fe sufficient conditions the transporter/s is/are downregulated, then treatments with hydroponic Fe would have less Cd, and conversely, treatments without Fe would increase the Cd uptake (because of the increased expression of the transporter/s). How would you explain this apparent discrepancies?
